# Substance Use among Street-Connected Children and Adolescents in Ghana and South Africa: A Cross-Country Comparison Study

**DOI:** 10.3390/bs11030028

**Published:** 2021-02-27

**Authors:** Kwaku Oppong Asante, Mashudu Tshifaro Nefale

**Affiliations:** 1Department of Psychology, University of Ghana, Legon, Accra, Ghana; 2Department of Psychology, University of the Free State, Bloemfontein 9301, South Africa; 3Department of Correctional Services, Pretoria 0001, South Africa; Mashudu.netshiombo@gmail.com

**Keywords:** homeless children, cross-national comparison, substance use, Ghana, South Africa

## Abstract

Using two cross-sectional surveys with a purposive sample of 376 homeless children and adolescents in both Ghana and South Africa, this study was conducted to examine the prevalence, sociodemographic and psychosocial correlates of substance use among street children and adolescents. An interviewer-administered questionnaire was used to obtain information on substance use, sociodemographic and psychosocial variables. Both bivariate and multivariate analyses showed that street-connected adolescents in Ghana reported higher prevalence of lifetime alcohol use and past-month alcohol use than those in South Africa. The protective effect of male gender was not observed in South Africa but significantly more pronounced in Ghana for all substances except past-month marijuana use. Sexual assault, indirect sexual victimization, physical beating, robbery, assault with a weapon and survival sex increased the odds of lifetime alcohol use and past-month alcohol use in Ghana. However, in South Africa, only robbery and assault with a weapon increased the odds of lifetime alcohol use while robbery and sexual assault increased the odds of past-month alcohol use. These results have implications for the development of harm reduction interventions, taking into consideration both the psychosocial and cultural context of substance use.

## 1. Introduction

The streets throughout the world are home to millions of children [1], and the circumstances on the street render these children and youth vulnerable to various kinds of psychological problems and health risks such as a high rate of sex trade [2,3,4] and substance abuse and misuse [5,6,7,8]. These health risk behaviors eventually put them at elevated risk for physical and mental health problems including sexually transmitted infections (STIs) [9,10,11].

There is global concern about the high prevalence of substance use among homeless children and adolescents [7,11,12]. However, most of the data on substance use and misuse is derived from studies conducted in economically advantaged and Western countries, particularly Canada, Australia and the United States. Although some evidence on the prevalence of substance use among homeless youth does exist in sub-Saharan Africa [2,9,10,13,14], studies with a cross-country comparison of the prevalence of substance use within the sub-region remain few. It may therefore be very useful to compare prevalence rates and correlates of substance use in a developed country with a vast research base such as that of South Africa, with rates and correlates in a developing country such as Ghana, which is currently undergoing significant economic and social transition. In addition, comparison between Ghana and South Africa may be cogent because of the numerous similar social and economic experiences of adolescents and youth in these countries. For example, just as migration and rapid urbanization, poverty, disintegrated families and divorce in Ghana force children and youth to leave their families prematurely to live and work on the streets [15,16], the freedom of choice and rapid economic development in South Africa may lead to public health challenges, including higher rates of substance use and orphans due to the death of a single or both parents [17,18]. Again, a high population of adolescents and young adults (between the ages of 15–24) and increase in unstable family relations are key social issues currently facing both countries [19,20,21]. Given that South Africa has developed programs and interventions for homeless children and adolescents, an examination of the South African experience may inform interventions that would aim at reducing homelessness and getting these vulnerable children and adolescents in Ghana off the street, where they are subjected to much trauma and abuse [2,14].

Cross-national comparisons are possible when common survey methods and measures are used. We compared the prevalence and correlates of substance use among Ghanaian and South African street children and adolescents between the ages of 8 and 19 years. Our aims and objectives were twofold. First, to compare the rates of past-month alcohol use, past-month cigarette use, past-month marijuana use and lifetime illegal drug use among homeless children and adolescents in Ghana to those among homeless children in South Africa. The second objective was to compare associations in the two countries between substance use and selected sociodemographic factors (i.e., age, gender), homelessness-related factors (i.e., length and reason for homelessness), homeless lifestyle risk factors (i.e., commercial or survival sex), and traumatic experiences (i.e., sexual assault, physical beating, robbery, assault with a weapon, indirect victimization) that have been shown to be related to substance use among homeless adolescents and youth [9,10,14,15,16].

## 2. Methods

### 2.1. Research Setting

The study was conducted in southern Ghana, specifically in the Greater Accra region, the smallest of the ten administrative regions in Ghana. It occupies a land area size of 3245 square kilometers and has a population of 4.0 million people (approximately 16% of the general Ghanaian population) [22]. As the capital city of Ghana, government ministries, departments and agencies, corporate headquarters of international and financial institutions as well as non-governmental organizations (NGOs) are located in Accra. The study was conducted among homeless children and adolescents in the CBD of Accra, where the second-largest number of street children in Ghana can be found [23]. In South Africa, the study was conducted in Durban, a city on the east coast in the province of KwaZulu-Natal. With regards to race, the Durban metropolitan area is predominantly inhabited by black South Africans (i.e., 74% of the total population in the Metropolis) [24]. The dominant home language is IsiZulu, spoken by around 62% of the population, followed by English at 26% [24]. Approximately 22% of the 10.2 million people living in the KwaZulu-Natal province are children aged 15 years or younger [24].

### 2.2. Participants and Procedure

The Ghanaian data were obtained from a sample of 227 homeless children (122 male and 105 female) aged 8–19 years (mean = 12.58, *SD* = 2.51), selected from the central business district (CBD) of Accra. The South African data were obtained from 149 street children (128 male and 21 female), aged 8–19 years (mean = 13.85, *SD* = 1.32), sampled from Durban, KwaZulu-Natal province. In both countries, a non-probability purposive sampling technique was used due to easier access to the street children population. An interviewer-administered questionnaire (due to the low level of education) was used for data collection. The majority of the participants in Ghana were interviewed in Twi and Ga (two predominant local languages spoken in Accra, Ghana), while those in South Africa were interviewed in either English or IsiZulu. Each participant was provided with some refreshment at the beginning of the interview as a token of appreciation for their willingness to participate in the study. Whereas the South African study sampled youth from the both NGO-housed accommodation and streets, the Ghanaian sample was drawn from only streets (due to difficulties in securing partnerships with NGO-catered accommodation and drop-in centers).

### 2.3. Ethical Consideration

Ethical approval to conduct the study was granted by the Department of Social Welfare, Accra, in Ghana and the Human and Social Science Ethics Committee (Ethical Approval number: HSS/0958/012) of the University of KwaZulu-Natal, Durban, South Africa. Due to their low educational background and inability to read, verbal consent was obtained from participants. The researcher read and translated the consent form into the preferred language of the participant. Participants were also informed that their participation in the study was voluntary and thus they had the right to withdraw from the study at any stage if they wished without any consequences to them. None of the participants expressed the need for psychological service although they were told of the availability of a psychologist should they require such a service. Confidentiality was maintained as participants were assured that the results of the study would be used for academic purposes only.

## 3. Measures

The survey questionnaires used in both countries were modeled on the South African Youth Risk Behavior Survey [18]. The development was also informed by a pilot study conducted among 52 homeless adolescents in similar but separate locations in both countries. This was done to determine the appropriateness of the questions asked and to ensure that it elicited relevant responses. However, some modifications were required. Some questions were adapted to local language and culture, for example, marijuana was referred to as “dagga” in South Africa and “weed” in Ghana. The items used to measure the various variables in the study are valid and reliable for use in both Ghana and South Africa [18,25].

*Demographics and homelessness-related factors****:*** Basic demographic characteristics used in the study included gender and age. We classified age into (3) age groups (8–10 years, 11–14 years and 15 years and older) which were created to ensure equal variance between the ages. These categorizations closely correspond to late childhood, early and late adolescence. These stages of development have distinctive features and abilities [25,26]. Two main homelessness-related factors were assessed: length and reason for homelessness. We categorized the reasons for being homeless into four (4) main groups, namely family poverty, dysfunctional family relations/divorce, physical abuse and sexual abuse. Race was excluded from the analysis due to the distinct categories of people from both countries. Length of homelessness was split into 3 categories: less than 1 year, 1–2 years and 3 years and more.

*Substance use:* For this study, substance use refers to the use of alcohol, marijuana, smoking of cigarettes and other drugs such as amphetamine, heroin (sugars) and glue. The questions on substance use included: *lifetime alcohol use:* “Have you ever drunk an alcoholic beverage?”; *past-month alcohol use:* “Have you used alcohol in the past month?”; *past-month cigarette use:* “Have you ever smoked a cigarette in the past month?”; *past-month marijuana use:* “Have you used marijuana in the past month?”; and *lifetime hard drugs use:* “Have you ever used any drugs such as amphetamine, glue or heroin?”. The response format to these items on the questionnaire was in the form of yes = 1 and no = 0. Due to a low response on hard drug use, we aggregated lifetime use of amphetamine, heroin (sugars) and glue into a composite variable of lifetime hard drug use.

*Homeless-lifestyle risk factors:* Homeless lifestyle risk factors were assessed with one variable: illegal income generating activities such as prostitution or survival sex. With survival sex, participants were asked whether in the past one month they had sex with someone in exchange for food, money and clothes or even a place to sleep. The response format to this question was in the form of yes = 1 and no = 0.

*Traumatic risk factors:* Traumatic risk factors were derived from 5 categorical questions which asked participants whether they had experienced any of these unpleasant events in the past 30 days: sexual assault, physical beating, robbery, assault with a weapon or indirect victimization such as witnessing someone being sexually assaulted. The response format to these questions was in the form of yes = 1 and no = 0.

## 4. Data Analyses

The Statistical Package for the Social Sciences (SPSS) version 26 for Windows (IBM SPSS, Inc., Chicago, IL, USA) was used to analyse the merged data. Substance use prevalence was calculated for the entire sample, by country and according to country sociodemographic characteristics. Odds ratio (OR) and 95% confidence interval (CI) were computed to compare relative prevalence rates between the 2 countries, with the Ghanaian sample serving as the reference group. Logistic regression was computed where each dependent variable (i.e., lifetime and past month alcohol use, past-month cigarette use, past-month marijuana use and lifetime hard drug use) was regressed into a set of explanatory variables to compute the adjusted odds ratio. All the variables selected for inclusion in the analysis were independent of each other, and thus, there was not a problem with multicollinearity. All analyses were two-tailed, and a *p*-value of less than 0.05 was considered statistically significant.

## 5. Results

### 5.1. Prevalence of Substance Use

The prevalence rates of lifetime alcohol use, past-month alcohol use, past-month cigarette use, past-month marijuana use and lifetime hard drug use in each of the countries are presented in Table 1. In South Africa, prevalence rates of lifetime alcohol use were significantly higher among males; in Ghana, the opposite pattern was observed. In Ghana, past-month cigarette use and past-month marijuana use were significantly higher among females but the opposite was observed for lifetime hard drug use. In the South African sample, no significant difference was observed in the prevalence rates of the other substances used by homeless adolescents.

The results further show that in the Ghanaian sample, age of participants was associated with all the substances used except for past-month cigarette use. There were no age variations in all the substances used among the South African sample. With the exception of lifetime hard drug use among the Ghanaian sample, reason for homelessness was not significantly associated with prevalence rates of other substances used in both countries.

In Ghana, the prevalence rates of lifetime alcohol use, past-month alcohol use, past-month cigarette use and past-month marijuana use increased with length of time of being homeless. There were variations in the prevalence rates of past-month marijuana use and lifetime hard drug use with length of time of being homeless in South Africa, but no specific pattern was observed.

Whereas all the traumatic risk factors (i.e., sexual assault, indirect sexual victimization, physical beating, robbery and assault with a weapon) were independently associated with alcohol use in Ghana, only robbery and assault with a weapon were related to alcohol use in South Africa. All the traumatic risk factors except physical beating were associated with past-month alcohol use and cigarette use in Ghana, while in South Africa, sexual assault and robbery were related to past-month alcohol use. No association was observed between the traumatic risk factors and past-month cigarette use. All the traumatic risk factors except for physical beating were associated with past-month marijuana use in Ghana but only robbery was related to past-month marijuana use in South Africa. Similarly, while sexual assault, robbery and assault with a weapon were related to lifetime hard drug use in Ghana, only robbery was found to be related to lifetime hard drug use in South Africa.

As reported in Table 1, survival sex was not associated with lifetime alcohol use and lifetime hard drug use in both Ghana and South Africa but had a significant association with past-month alcohol use and past-month cigarette use only in Ghana.

### 5.2. Cross-Country Comparisons of Correlates

The inter-country odds ratios are presented in Table 2. The protective effect of male gender was significantly more pronounced in Ghana for all substance use measures except past-month marijuana use, but such gender effect was not observed in South Africa. In Ghana, participants aged 11–14 years and 15 years and above were more likely to have lifetime and past-month alcohol use, but those aged 15 years and above were found to be less likely to have a lifetime of hard drugs use. In South Africa, participants aged 11–14 years were at greater odds of past-month marijuana use.

In Ghana, the odds of having used all the various substances were very high for participants who had been homeless for 1–2 years. However, homeless adolescents who had lived on the street for three years or more were less likely to have engaged in both lifetime and past-month alcohol use. No such effect was observed in South Africa. Sexual assault, indirect sexual victimization, physical beating, robbery, assault with a weapon and survival sex increased the odds of alcohol use and past-month alcohol use in Ghana but only robbery and assault with a weapon increased the odds of alcohol use, while robbery and sexual assault increased the odds of past-month alcohol use in South Africa.

Similarly, sexual assault, indirect sexual victimization, robbery, assault with a weapon and survival sex increased the odds of past-month cigarette use in Ghana, but no such effect was observed among the South African sample.

Our results further showed that sexual assault, indirect sexual victimization, robbery, assault with a weapon and survival sex increased the odds of past-month marijuana use in Ghana but only robbery was found to increase the odds of past-month marijuana use in South Africa. In contrast, participants in Ghana who had ever been robbed and assaulted with a weapon were less likely to have had lifetime hard drug use. No such effect was observed among the South African sample.

## 6. Discussion

This study was conducted to examine the prevalence and psychosocial correlates of substance use among Ghanaian and South African street-connected children and adolescents. The results indicated that lifetime and past-month alcohol use was higher among Ghanaian than among South African street children and adolescents, but rates of past-month marijuana use and lifetime hard drug use were lower. High prevalence of alcohol use among Ghanaian street children and adolescents has been reported elsewhere and may appear to be related to easy accessibility to both minors and adults [27,28]. The relatively high rates of marijuana and lifetime hard drug use as found among South African street children and adolescents were consistent with data from other developed countries [7,29], and lower rates reported among Ghanaian street children and adolescents were consistent with data from other sub-Saharan African countries [2,11,28].

The legal age limit for tobacco use is 18 years for individuals in Ghana and South Africa. However, our results show that past-month cigarette use was appreciably higher in South Africa than in Ghana. These differences could be attributed to the conservative culture in Ghana where the use of tobacco is highly stigmatized [30]. It is socially unacceptable for individuals to smoke tobacco in Ghana, as it has been in many developing countries [31,32]. It is thus, possible, that stigmatization within such cultures may serve as a way of socially controlling tobacco use.

The protective effect of male gender was significantly more pronounced in Ghana for all substances except past-month marijuana use, but such gender effect was not observed in South Africa. These findings contradict previous studies among homeless adolescents where males generally reported higher substance use than females [5,9]. The odds of females using more substances than males in Ghana could be attributable to the fact that homeless female adolescents with sexual abuse histories do abuse drugs on the streets [3], and their abuse of substances could be a coping mechanism for survival while being homeless [15]. Some street-connected children and adolescent also use drugs not only as adaptive strategies for the unpleasant conditions, but as a way to get accepted into the street culture and to be regarded with respect and fear [15]. Thus, in order to be accepted as someone on the streets/hustler you need to do what is considered the way of life. In South Africa, there has been a proliferation of street drugs including marijuana, cocaine, and heroin, and it is possible that these substances are available equally to both males and females, as a high percentage of these adolescents experiment with illegal mood-altering drugs at a young age, and with highly addictive drugs such as methamphetamine [33,34].

Sexual assault, indirect sexual victimization, physical beating, robbery, assault with a weapon and survival sex increased the odds of lifetime and past-month alcohol use in Ghana but only robbery and assault with a weapon increased the odds of lifetime alcohol use, while robbery and sexual assault increased the odds of past-month alcohol use in South Africa. It was also observed that sexual assault, indirect sexual victimization, robbery, assault with a weapon and survival sex increased the odds of past-month marijuana use in Ghana but only robbery was found to increase the odds of past-month marijuana use in South Africa. On the contrary, participants in Ghana who had ever been robbed and assaulted with a weapon were less likely to have had lifetime hard drug use. No such effect was observed in the South African sample. These findings reported from both countries on the relationship between substance use and violence and violence-related behaviors can be explained from two different perspectives. Firstly, these findings support the clustering effect of health risk behaviors among this vulnerable population who are predisposed to several risk factors heightening multiple vulnerabilities [2,12]. Previous studies conducted within sub-Saharan Africa (SSA) have reported the presence of co-occurrence of health-compromising behaviors among street children and adolescents in countries such as Egypt, Malawi and Sudan [2,35,36]. It is thus plausible that substance use may be viewed as a risk factor related to other health-damaging behaviors, particularly violence and violence-related behaviors.

Secondly, the relationship between alcohol use and violence-related behaviors reported in both countries could be attributed to street adolescents’ use of various substances within the street environment as a coping strategy to deal with level of aggression that is required to survive the highly competitive street environment. Substance use has been found to be strongly entrenched in street life culture and employed to manage the multitude of challenges they face [7,11,12,13,14,15,37]. These avoidance coping behaviors are likely to be facilitated by homeless adolescents’ inability to access psychological and health care services and having inadequate knowledge and skills regarding appropriate coping strategies.

The study has two major strengths. First, this is one of the first studies particularly within sub-Saharan Africa that compares the prevalence and correlates of substance use between a developed country with a vast research base such as that of South Africa, and a developing country such as Ghana. Secondly, the sample size used was fairly large, thus increasing the generalizability of the findings. Despite these strengths, there are acknowledged limitations. Measures were based on self-reports, which might have been confounded by systemic bias and social desirability. Our study may have also been influenced by different levels of reliability and validity in both study sites. However, in the administration of the research instrument in both countries, steps were taken to reduce such biases, and participation was voluntary and utmost confidentiality was assured. Direct comparison of the current results with other studies in other areas should be noted with caution because variations reported between Ghana and South Africa may be due to environmental and methodological variations rather than the noted differences per se. Finally, key psychosocial factors (e.g., depression, social support, stress) which are known to be associated with substance use among homeless children and adolescents were not included. Further cross-national comparison studies would benefit from the inclusion of broader risk and protective factors.

## 7. Conclusions

This study was conducted to examine the prevalence, as well as sociodemographic and psychosocial correlates of substance use among Ghanaian and South African street children and adolescents. Our results showed that street-connected adolescents in Ghana reported higher prevalence of alcohol and past-month alcohol use than those in South Africa. The protective effect of male gender was significantly more pronounced in Ghana for all substances except past-month marijuana use, but such gender effect was not observed in South Africa. Sexual assault, indirect sexual victimization, physical beating, robbery, assault with a weapon and survival sex increased the odds of alcohol use and past-month alcohol use in Ghana. However, in South Africa, only robbery and assault with a weapon increased the odds of alcohol use while robbery and sexual assault increased the odds of past-month alcohol use. These results have implications for the development of harm reduction interventions, taking into consideration both psychosocial and the cultural context of substance use. Additionally, substance use cessation strategies (e.g., both cognitive and behavioral strategies) could be instituted while regular education campaigns on health-compromising behaviors could also be employed. These interventions are particularly important as the coronavirus (COVID-19) global pandemic exacerbates the vulnerability of this population to drug use.

## Figures and Tables

**Table 1 behavsci-11-00028-t001:** Substance use prevalence rates among homeless children by sociodemographic and psychosocial characteristics: Ghana and South Africa.

Health Behavior	Lifetime Used Alcohol, % (95%CI)		Past-Month Alcohol Use, % (95%CI)		Past-Month Cigarette Use, %(95%CI)		Past-Month Marijuana Use, %(95%CI)		Lifetime Hard Drug Use, % (95%CI)	
	Ghana	South Africa	Ghana	South Africa	Ghana	South Africa	Ghana	South Africa	Ghana	South Africa
**Gender**										
Male	82.1(73.7, 89.5) **	87.3(81.0, 92.9) *	74.7(65.3, 84.2)	74.6(66.7, 82.5)	68.4(58.9, 76.8) *	99.2(96.8, 99.8)	68.4(59.8, 76.5) *	93.7(88.9, 97.6)	24.2(15.8, 33.7) **	45.0(36.0, 54.1)
Female	91.1(83.6, 97.5)	71.4(47.6, 90.5)	75.9(65.8, 84.8)	71.4(52.4, 90.5)	74.7(64.6, 83.5)	85.3(83.4, 89.2)	81.0(72.2, 88.6)	90.3(76.2, 98.5)	13.9(6.3, 22.8)	55.0(30.0, 75.0)
**Age**										
8–10 years	63.9(47.2. 77.8) ***	82.9(68.2, 94.3)	61.1(44.4, 75.0) **	71.4(54.4, 85.7)	58.3(41.7, 72.2)	98.0(97.0, 99.8	58.3(41.7, 72.1) *	85.7(74.3, 97.1)	44.4(27.8, 61.1) ***	51.5(36.4, 69.7)
11–14 years	92.4(86.7, 97.1)	90.4(80.0, 98.0)	82.9(75.3, 89.5)	80.0(66.7, 93.3)	72.4(63.8, 81.0)	96.7 (90.0, 99.0)	81.0(73.3, 88.6)	93.3(83.3, 99.0)	11.4(5.7, 17.1)	42.3(23.1, 61.5)
15 years and over	96.8(90.3, 98.9)	84.1(75.6, 91.5)	71.1(54.8, 87.1)	73.3(63.4, 81.7)	83.9(71.0, 93.5)	98.5(96.0, 99.8)	71.0(54.8, 87.1)	96.3(91.5, 99.8)	19.4(6.5, 32.3)	42.6(30.9, 54.4)
**Reasons for homelessness**										
Family poverty	84.2(76.2, 90.1)	89.2(78.4, 97.3)	71.3(61.4, 79.2)	81.1(67.6, 91.9)	73.3(64.4, 82.2)	83.3(74.4, 85.8)	73.3(64.4, 81.2)	94.6(86.5, 99.8)	20.8(12.9, 28.7) *	48.6(32.4, 64.9)
Divorce	85.2(70.4, 96.3)	89.7(75.9, 99.8)	81.5(66.7, 96.3)	82.8(65.5, 96.6)	66.7(48.1, 85.2)	55.8(44.2, 79.8)	77.8(63.0, 92.6)	89.7(79.3, 98.9)	22.2(7.4, 40.7)	51.7(34.5, 69.0)
Sexual abuse	90.0(75.0, 98.9)	88.9(72.2, 90.8)	80.0(60.0, 95.0)	77.8(61.1, 94.4)	70.0(50.0, 89.9)	82.0(72.5, 90.1)	85.0(70.0, 98.9)	94.4(83.3, 99.8)	15.0(13.5, 34.9)	50.0(27.8, 72.2)
Physical abuse	91.3(78.3, 99.4)	80.0(66.7, 91.1)	87.0, 73.9, 96.8)	68.9(55.6, 80.0)	78.3(60.9, 95.7)	88.3(70.7, 90.1)	73.9(56.5, 91.3)	93.3(84.4, 99.8)	17.414.3, 34.8)	40.0(24.4, 53.3)
**Length of time of being homeless**										
<1 year	60.0(33.3, 86.7) ***	85.7(71.5, 97.1)	53.3(26.7, 80.0) **	78.4(66.7, 90.1)	26.7(6.8, 46.7) ***	70.6(56.9, 82.4)	53.4(26.7, 80.1) **	96.0(94.1, 98.0) *	26.7(6.7, 53.3)	34.9(20.9, 48.8) *
1–2 years	81.0(72.2, 89.9)	85.4(75.0, 93.8)	73.4(63.3, 83.5)	89.6(79.2, 95.8)	72.2(62.0, 82.3)	77.1(64.6, 89.5)	74.7(65.8, 83.5)	85.4(75.0, 93.8)	20.3(12.7, 29.1)	60.5(46.5, 74.4)
3 years and more	96.1(90.9, 98.5)	82.9(70.7, 92.7)	83.1(74.0, 90.9)	88.4(79.1, 97.7)	81.3(72.7, 90.9)	76.7(62.8, 88.4)	80.5(71.4, 88.3)	97.8(93.0, 99.8)	18.2(10.4, 27.3)	40.0(26.2, 54.8)
**Traumatic risk factors**										
Sexual assault	96.4(91.9, 98.1) ***	90.6(80.0, 98.9)	89.3(80.4, 96.4) **	90.6(83.3, 99.5) *	78.6(67.9, 89.3) *	98.0(88.4, 99.3)	87.5(78.6, 94.6) ***	96.9(90.0, 98.9)	33.9(23.2, 46.4) **	33.3(16.7, 50.0)
Indirect sexual victimization	94.2(86.0, 98.5) **	98.2(92.0, 99.8)	90.4(90.8, 96.4) ***	84.2(71.4, 94.3)	92.2(84.6, 98.2) ***	97.0(92.0, 98.5)	94.2(86.5, 98.8) ***	94.7(85.7, 98.0)	25.0(13.5, 36.5)	48.6(31.4, 65.7)
Physical beating	85.2(79.2, 91.3) ***	85.6(80.3, 92.6)	76.5(69.1, 83.2)	76.3 (71.3, 86.0)	70.5(63.1, 77.9)	99.3(97.5, 99.8)	73.8(66.4, 81.2)	92.8(87.7, 96.7)	17.4(11.4, 24.1)	47.2(38.5, 57.4)
Robbery	92.1(87.1, 96.4) ***	96.4(88.3, 98.6) **	81.4(75.0, 87.9) ***	84.2(77.2, 87.9) **	75.7(68.6, 82.9) ***	77.2(66.8, 85.2)	80.0(73.6, 86.4) **	88.9(86.7, 95.9) **	15.7(10.0, 22.1) ***	35.2(29.8, 44.3) ***
Assaulted with a weapon	94.2(90.0, 98.3) ***	95.0(92.2, 97.8) **	89.2(83.3, 94.2) ***	80.9(75.0, 91.2)	85.0(78.3, 90.5) ***	86.0(70.2, 88.5)	89.3(83.3, 94.2) ***	90.8(88.9, 98.2)	15.8(9.2, 22.5) ***	45.2(31.2, 52.0)
**Illegal income generating**										
Survival sex	95.9(91.8, 99.0) ***	76.5(52.9, 94.1)	89.8(83.7, 94.9) ***	82.4(64.7, 98.0)	84.7(77.6, 91.8) ***	98.1(90.1, 99.8)	92.9(87.8, 96.9) ***	94.1(82.4, 99.0)	19.4(12.2, 27.6)	41.2(17.6, 54.7)

Note: * *p* < 0.05; ** *p* < 0.01; *** *p* < 0.001.

**Table 2 behavsci-11-00028-t002:** Odds ratio and their 95% confidence interval of sociodemographic and psychosocial characteristics predictors of substance use in Ghana and South Africa.

Health Behavior	Lifetime Used Alcohol, OR (95%CI)		Past-Month Alcohol Use, OR (95%CI)		Past-Month Cigarette Use, OR (95%CI)		Past-Month Marijuana Use, OR (95%CI)		Lifetime Hard Drug Use, OR (95%CI)	
	Ghana	South Africa	Ghana	South Africa	Ghana	South Africa	Ghana	South Africa	Ghana	South Africa
**Gender**										
Male	1	1	1	1	1	1	1		1	1
Female	3.2(1.5, 5.0) **	0.4(0.1, 1.1)	1.5(0.8, 2.8)	0.8(0.3, 2.4)	1.8(1.2, 3.3) *	1.2(0.3, 6.1)	2.0(1.1, 3.7) *	0.6(0.1, 3.3)	0.4(0.2, 0.8) *	1.2(0.8, 8.8)
**Age**										
8–10 years	1	1	1	1	1	1	1	1	1	1
11–14 years	3.9(1.3, 6.2) *	1.1(0.4, 3.2)	1.5(0.6, 3.5)	1.4(0.5, 3.5)	1.4(0.6, 3.4)	1.3(0.8, 4.8)	2.2(0.8, 5.7)	4.4(1.9, 8.5) *	0.5(0.2, 1.3)	0.8(0.5, 3.3)
15 years and over	1.9(0.7, 5.2)	0.6(0.2, 2.3)	2.8(1.3, 6.2) **	0.9(0.4, 2.5)	1.2(0.6, 2.5)	0.4(0.2, 1.8)	1.2(0.5, 2.8)	1.9(0.3, 11.8)	0.3(0.1, 0.7) **	0.4(0.2, 2.7)
**Reasons for homelessness**										
Family poverty	1.2(0.4, 4.1)	0.5(0.1, 1.8)	0.8(0.3, 2.6)	0.4(0.1, 1.4)	1.7(0.6, 4.9)	2.9(0.9, 4.9)	0.8(0.3, 2.4)	1.6(0.3, 8.3)	0.7(0.2, 2.3)	1.7(0.91, 5.3)
Divorce	1	1	1	1	1	1	1	1	1	1
Sexual abuse	1.2(0.4, 3.1)	0.4(0.1, 1.7)	1.5(0.6, 3.8)	0.5(0.2, 1.5)	1.5(0.6, 3.8)	1.8(0.8, 2.4)	0.7(0.3, 1.8)	0.8(0.1, 5.1)	0.6(0.2, 1.7)	0.9(0.4, 4.7)
Physical abuse	0.4(0.1, 2.3)	0.8(0.3, 2.8)	0.6(0.2, 2.3)	1.3(0.4, 3.2)	1.6(0.5, 5.2)	0.4(0.2, 3.1)	0.5(0.1, 1.7)	0.9(0.2, 5.6)	1.2(0.3, 5.9)	2.1(0.6, 3.0)
**Length of time of being homeless**										
<1 year	1	1	1	1	1	1	1	1	1	1
1–2 years	12.1(3.8, 18.4) ***	2.1(0.7, 6.6)	6.5(2.7, 10.8) ***	1.4(0.5, 3.5)	8.7(3.2, 14.0) ***	1.4(0.4, 2.5)	5.20(1.9, 13.9) ***	0.8(0.1, 10.3)	0.5(0.2, 1.4)	1.5(0.8 3.9)
3 years and more	0.2(0.1, 0.6) ***	0.9(0.2. 3.2)	0.5(0.2, 0.9) **	1.0(0.4, 2.5)	0.6(0.3, 1.1)	0.6(0.2, 1.8)	0.9(0.4, 1.7)	4.2(0.8, 14.5)	0.5(0.7, 2.9)	0.8(0.5, 4.1)
**Traumatic risk factors**										
Sexual assault	4.5(1.2, 7.6) *	1.9(0.5, 6.8)	4.0(1.8, 9.0) ***	4.2(1.2, 14.6) *	2.2(1.3, 4.3) *	1.8(0.3, 3.1)	4.3(1.9, 7.8) ***	2.6(0.3, 11.6)	2.4(1.2, 4.6) *	1.2(0.2, 3.1)
Indirect sexual victimization	6.4(1.9, 11.8) **	1.4(0.2, 3.8)	6.4(2.4, 10.9) ***	2.2(0.8, 5.7)	9.9(3.4, 11.8) ***	2.2(0.5, 6.4)	11.5(3.4, 18.7) ***	1.4(0.2, 7.0)	1.2(0.6, 2.4)	0.9(0.4, 1.5)
Physical beating	0.2(0.1, 1.5)	1.7(0.3, 8.8)	0.8(0.3, 2.3)	4.0(0.8, 5.8)	0.4(0.1, 1.4)	1.2(0.8, 3.3)	0.4(0.1, 1.4)	0.4(0.2, 2.4)	0.5(0.2, 1.2)	0.2(0.1, 4.4)
Robbery	10.0(4.5, 12.2) **	5.8(1.2, 10.2) **	5.80(2.7, 12.2) ***	2.9(1.4, 8.2) **	4.0(1.9, 8.1) ***	0.3(0.1, 1.3)	3.7(1.7, 8.2) ***	7.3(2.8, 10.7) ***	0.2(0.1, 0.3) ***	0.6(0.3,2.3)
Assaulted with a weapon	10.4(4.7, 13.1) **	6.4(2.5, 8.9) **	12.9(6.3, 16.7) ***	2.1(0.6, 4.8)	8.3(4.3, 11.2) ***	2.1(0.8, 4.8)	9.6(4.7, 13.3) ***	2.3(0.8, 8.5)	0.3(0.2, 0.6) **	0.5(0.1, 0.8)
**Illegal income generating**										
Survival sex	9.4(2.9, 13.6) ***	0.6(0.2, 2.1)	8.8(3.4, 13.9) ***	1.3(0.4, 4.2)	3.2(1.8, 7.6) ***	0.8(0.2, 1.8)	7.53(3.4, 13.9) **	1.2(0.1, 10.3)	0.6(0.3, 1.1)	0.8(0.4, 2.0)

Note: * *p* < 0.05; ** *p* < 0.01; *** *p* < 0.001.

## Data Availability

All data generated during and/or analyzed during the study are available from the corresponding author on reasonable request.

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
