# Peer review of "Substance Use among Street-Connected Children and Adolescents in Ghana and South Africa: A Cross-Country Comparison Study"

_behavsci, 2021, doi:10.3390/bs11030028_

Round 1

Reviewer 1 Report

The study is very important because the problem of homelessness, especially for children, is a serious and definitely a situation   at can be addressed.

Abstract: it should be clarified what has been meant by “socio-demographic and psychosocial correlates…” (Row 12), one might imagine that these are socio-demographic data, but what is below psychosocial is difficult to imagine because the range is so huge.

Introduction part has been well structured, revealing the topicality of the study. At the end, a description of the objectives has been provided, which consisted of two sub-objectives.

Methodology part: I would like to understand the sentence: "The Durban metro is predominantly black African (74%) with colored in the minority at 3% [22]." (Row 75).

Participants part - in the abstract it was shown that 345 children and adolescents were interviewed, but in this section 227 and 149 were presented, which makes up 376 respondents.

Substance use part - “other drugs such as amphetamine heroin (sugars)” (Row 129) - does such a mentioning mean that the substance contains both amphetamine and heroin? Is it a new substance and common to both countries? This should be explained why such an offer to respondents. Or is it a typo that needs to be corrected?

Result part: The authors should a little bit clarify the meaning of sentences, for example, "However, being homeless for 3 years or more was protective of both lifetime and past-month alcohol use." (Row 214) What does "protective" mean in this context?

Discussion part - The authors could provide more justification for substance use among female adolescents in Ghana compared to South Africa.

Author Response

The study is very important because the problem of homelessness, especially for children, is a serious and definitely a situation   at can be addressed.

Response: Thank you for this comment.

Abstract: it should be clarified what has been meant by “socio-demographic and psychosocial correlates…” (Row 12), one might imagine that these are socio-demographic data, but what is below psychosocial is difficult to imagine because the range is so huge.

Response: Generally, our study aimed at looking at socio-demographic data (i.e. age, gender and reason for being homeless) while psychosocial factors refers to traumatic risk factors and engagement in survival sex (please refer to Table 1)

Introduction part has been well structured, revealing the topicality of the study. At the end, a description of the objectives has been provided, which consisted of two sub-objectives.

Response: Thank you.

Methodology part: I would like to understand the sentence: "The Durban metro is predominantly black African (74%) with colored in the minority at 3% [22]." (Row 75).

Response: We have now re-structured the sentence to read “With regards to race, the Durban metropolitan area  is predominantly inhabited by black South African (i.e. 74% of the total population in the Metropolis).

Participants part - in the abstract it was shown that 345 children and adolescents were interviewed, but in this section 227 and 149 were presented, which makes up 376 respondents.

Response: Thank you. We have now corrected this typographical error. Refer to line 12 in the abstract.

Substance use part - “other drugs such as amphetamine heroin (sugars)” (Row 129) - does such a mentioning mean that the substance contains both amphetamine and heroin? Is it a new substance and common to both countries? This should be explained why such an offer to respondents. Or is it a typo that needs to be corrected?

Response: Thank you. This was a typo error. It should have been “……other drugs such as amphetamine, heroin (sugars) and glue.

Result part: The authors should a little bit clarify the meaning of sentences, for example, "However, being homeless for 3 years or more was protective of both lifetime and past-month alcohol use." (Row 214) What does "protective" mean in this context?

Response: The sentence has now been changed to read “ However, homeless adolescents who have lived on the street for 3 years or more were less likely  to have engaged in both lifetime and past-month alcohol use. See line 215-216

Discussion part - The authors could provide more justification for substance use among female adolescents in Ghana compared to South Africa.

Response: Our data and the way the comparison was done, may not give us the opportunity to do such comparison.

Reviewer 2 Report

The paper is very well prepared and deals with a topic marginalised in studies written from the perspective of highly developed countries. Due to the specificity of the population, the risk factors analysed are also unique. 

I suggest just a few changes, clarifications or extensions.

Line 36 - the European region cannot be overlooked. Europe has a number of systems for monitoring the health behaviour of adolescents, but by definition, only young people attending school are covered.  This is worth mentioning: HBSC (Health Behaviour in School-aged Children www.hbsc.org , ESPAD ( European School Survey Project on Alcohol and Other Drugs www.espad.org , GYTS (Global Youth Tobacco Survey www.who.int/tobacco/surveillance/gyts/en. Please add references, or at least web addresses. These systems not only provide information about the scale of the phenomenon, but also about potential conditions, giving the opportunity to follow trends and international comparisons.     

It would greatly enhance the value of the paper (and place it in a deeper context) to add a broader view of public health in the introduction, referring to the concept of universal, selective and indicated prevention. Homeless children are a remarkable example of a target population that should be covered by this third and most difficult level. I also suggest that reference be made to so called indicated prevention in the conclusions .

I expect homeless children as hard to reach group, it’s probably difficult to arrange an interview with them. This is a poorly explained aspect.

It also follows from the description that there are registered systems for the homeless in both countries. Is it also possible to expect a significant number of unregistered cases. That would be another limitation of the study. With such high rates of use of psychoactive substances, this probably does not interfere with the conclusions. 

The description of the methods of analysis focuses on giving an OR with a CI. It is worth noting that prevalence indicators were also given with CI. 

The description of items about the use of the substance is not accurate (means not consistent with tables) and focused on life time prevalence. It’s worth explaining that lifetime and past month prevalence were studied separately.

The tables are very clear. However, it would be great to add "total" as the first raw of Table 1. This would give the opportunity to know the overall frequency and compare with other previous and future studies.

References - please notice that in MDPI journals year of publication should be bolded. Please, check also journal abbreviations. Eg. #36 Behav Sci.

Author Response

The paper is very well prepared and deals with a topic marginalised in studies written from the perspective of highly developed countries. Due to the specificity of the population, the risk factors analysed are also unique. 

Response: Thank you

I suggest just a few changes, clarifications or extensions.

Line 36 - the European region cannot be overlooked. Europe has a number of systems for monitoring the health behaviour of adolescents, but by definition, only young people attending school are covered.  This is worth mentioning: HBSC (Health Behaviour in School-aged Children www.hbsc.org , ESPAD ( European School Survey Project on Alcohol and Other Drugs www.espad.org , GYTS (Global Youth Tobacco Survey www.who.int/tobacco/surveillance/gyts/en. Please add references, or at least web addresses. These systems not only provide information about the scale of the phenomenon, but also about potential conditions, giving the opportunity to follow trends and international comparisons.     

Response: I thank the reviewer for these recommendations of these key indicators. On the contrary, we of the opinion that these indicators may not be relevant for this manuscript. We have contextually situated our paper in an African context; thus reference to other indices in Europe may not be relevant.

It would greatly enhance the value of the paper (and place it in a deeper context) to add a broader view of public health in the introduction, referring to the concept of universal, selective and indicated prevention. Homeless children are a remarkable example of a target population that should be covered by this third and most difficult level. I also suggest that reference be made to so called indicated prevention in the conclusions.

Response: We have already done this and included in the conclusion some few implications of the study. See conclusion section.

I expect homeless children as hard to reach group, it’s probably difficult to arrange an interview with them. This is a poorly explained aspect.

Response: It is indeed true that homeless adolescents are hard to reach population. In this study, social workers in the two countries who had extensive working experience with homeless young adults through involvement with two non-governmental organizations were recruited to help with the identification of the participants. Their involvement helped bridged the gap between the researchers and the street children, so that the youth would feel comfortable in answering the answering of the items on the questionnaire.  

It also follows from the description that there are registered systems for the homeless in both countries. Is it also possible to expect a significant number of unregistered cases. That would be another limitation of the study. With such high rates of use of psychoactive substances, this probably does not interfere with the conclusions. 

Response: This comment is not clear. We are however of the opinion that since the sample use is limited to those who lived purely on the street, the findings may not be generalized to represent those who are being provided with housing.

The description of the methods of analysis focuses on giving an OR with a CI. It is worth noting that prevalence indicators were also given with CI. 

Response: Thank you. The presentation of the prevalence of substance use with their CI is a normal practice. This is done to give the reader of the possible range of the reported figure. This does not necessarily have to be stated explicitly in the analysis section as done for the OR.

The description of items about the use of the substance is not accurate (means not consistent with tables) and focused on lifetime prevalence. It’s worth explaining that lifetime and past month prevalence were studied separately.

Response: Thank you very much for this comment. We have now made sure that the names of the indices of substance use as presented in the methods are the same as those presented in the results and the various Tables. Please refer to lines 132-135

The tables are very clear. However, it would be great to add "total" as the first raw of Table 1. This would give the opportunity to know the overall frequency and compare with other previous and future studies.

Response: This is a good recommendation. However, the nature of the current tables makes it impossible to insert additional row for each of the indices of substance use for each country. That maybe too much to be fixed in the current tables.

References - please notice that in MDPI journals year of publication should be bolded. Please, check also journal abbreviations. Eg. #36 Behav Sci.

Response: This has now been corrected. Thank you

Reviewer 3 Report

Overall, this is a well-articulated study that flows seamlessly from section to section.

The authors argue from line 34 that there is a paucity of studies in developing countries. There are in fact some studies that have addressed this issue that would inform your theoretical and methodological section. I have provided three references below that should be incorporated into the paper.

Being that this paper is going to be published with the coronavirus global pandemic as a backdrop, it is important to highlight the vulnerability of the street-connected youth to the pandemic which is exacerbated by drug use. This could either be in the discussion or conclusion sections.

Recommended articles:

Yahya Muhammed Bah (2018) Drug Abuse among Street Children. Journal of Clinical Research In HIV AIDS And Prevention - 3(3):12-45.

DOI10.14302/issn.2324-7339.jcrhap-18-2291

Ayenew, M., Kabeta, T. & Woldemichael, K. Prevalence and factors associated with substance use among street children in Jimma town, Oromiya national regional state, Ethiopia: a community based cross-sectional study. Subst Abuse Treat Prev Policy 15, 61 (2020). https://doi.org/10.1186/s13011-020-00304-3

Cumber, S, & Tsoka-Gwegweni, J. (2016). Pattern and practice of psychoactive substance abuse and risky behaviours among street children in Cameroon. South African Journal of Child Health, 10(3), 166-170. https://dx.doi.org/10.7196/sajch.2016.v10i3.1066

Author Response

Overall, this is a well-articulated study that flows seamlessly from section to section.

Response: Thank you

The authors argue from line 34 that there is a paucity of studies in developing countries. There are in fact some studies that have addressed this issue that would inform your theoretical and methodological section. I have provided three references below that should be incorporated into the paper.

Response: These recommended articles are very helpful. I have inserted the 2 relevant ones to this manuscript. See reference # 13 and 14 and lines 38-40 in the introduction section.

Being that this paper is going to be published with the coronavirus global pandemic as a backdrop, it is important to highlight the vulnerability of the street-connected youth to the pandemic which is exacerbated by drug use. This could either be in the discussion or conclusion sections.

Response: I have inserted a little bit of the vulnerability of this population in times of COVID-19 into the conclusion section.

Recommended articles:

Yahya Muhammed Bah (2018) Drug Abuse among Street Children. Journal of Clinical Research In HIV AIDS And Prevention - 3(3):12-45. DOI10.14302/issn.2324-7339.jcrhap-18-2291

Ayenew, M., Kabeta, T. & Woldemichael, K. Prevalence and factors associated with substance use among street children in Jimma town, Oromiya national regional state, Ethiopia: a community based cross-sectional study. Subst Abuse Treat Prev Policy 15, 61 (2020). https://doi.org/10.1186/s13011-020-00304-3

Cumber, S, & Tsoka-Gwegweni, J. (2016). Pattern and practice of psychoactive substance abuse and risky behaviours among street children in Cameroon. South African Journal of Child Health, 10(3), 166-170. https://dx.doi.org/10.7196/sajch.2016.v10i3.1066

Response: These recommended articles are very helpful. I have inserted the 2 relevant ones to this manuscript. See reference # 13 and 14.